# A deeply supervised adaptable neural network for diagnosis and classification of Alzheimer's severity using multitask feature extraction

Mohsen Ahmadi [1], Danial Javaheri [2]*, Matin Khajavi [3], Kasra Danesh [1], Junbeom Hur [2]*

1 Department of Electrical Engineering and Computer Science, Florida Atlantic University, Boca Raton, FL, United States of America, 2 Department of Computer Science and Engineering, Korea University, Seoul, Republic of Korea, 3 Foster School of Businesses, University of Washington, Seattle, Washington, United States of America

* jbhur@korea.ac.kr (JH); javaheri@korea.ac.kr (DJ)

## Abstract

Alzheimer's disease is the most prevalent form of dementia, which is a gradual condition that begins with mild memory loss and progresses to difficulties communicating and responding to the environment. Recent advancements in neuroimaging techniques have resulted in large-scale multimodal neuroimaging data, leading to an increased interest in using deep learning for the early diagnosis and automated classification of Alzheimer's disease. This study uses machine learning (ML) methods to determine the severity level of Alzheimer's disease using MRI images, where the dataset consists of four levels of severity. A hybrid of 12 feature extraction methods is used to diagnose Alzheimer's disease severity, and six traditional machine learning methods are applied, including decision tree, K-nearest neighbor, linear discrimination analysis, Naïve Bayes, support vector machine, and ensemble learning methods. During training, optimization is performed to obtain the best solution for each classifier. Additionally, a CNN model is trained using a machine learning system algorithm to identify specific patterns. The accuracy of the Naïve Bayes, Support Vector Machines, K-nearest neighbor, Linear discrimination classifier, Decision tree, Ensembled learning, and presented CNN architecture are 67.5%, 72.3%, 74.5%, 65.6%, 62.4%, 73.8% and, 95.3%, respectively. Based on the results, the presented CNN approach outperforms other traditional machine learning methods to find Alzheimer severity.

## 1 Introduction

Alzheimer's disease is a type of dementia that affects the elderly. The onset of symptoms is estimated to occur 15–-20 years prior to the onset of the disease. Neurons important in memory, reasoning, and learning function are destroyed, resulting in syndromes [1]. The disease's specific aetiology and treatment are yet unclear. Researchers have used a variety of neuroimaging

**Data Availability Statement:** This paper used an online dataset. https://www.kaggle.com/datasets/tourist55/alzheimers-dataset-4-class-of-images.

**Funding:** This research was supported by Basic Science Research Program through the National Research Foundation of Korea (NRF) funded by the Ministry of Education (NRF-2021R1A6A1A13044830) and ICT Creative Consilience program funded by Ministry of Science and ICT (IITP-2023-2020-0-01819), Korea, supervised by the IITP (Institute for Information & communications Technology Planning & Evaluation).

**Competing interests:** The authors have declared that no competing interests exist.

methods, including single-photon emission computed tomography, magnetic resonance imaging (MRI), and positron emission tomography, to study AD [2]. In 2017, there were 121,404 deaths associated with AD, making it the sixth leading cause of death in the United States. It is estimated that 60 million individuals will be affected by Alzheimer's disease over the next 20 years. The World Alzheimer's Report estimates that there will be 152 million Alzheimer's patients by 2050 [3]. A scientist examines the thickness of the cortex, the density of gray matter, as well as the enlargement of the ventricles and the shrinkage of the brain. The cerebrospinal fluid (CSF), gray matter (GM), and white matter (WM) are the three primary tissues in brain imaging. However, researchers found that GM atrophy is more closely related to cognitive impairment in patients with MCI [4]. According to AD research, finding accurate biomarkers for the automated diagnosis of AD or MCI has been a promising and hard endeavor in recent years [5]. Many biomarker study initiatives have already begun and have shown significant findings.

The Alzheimer's Association and the National Institute on Aging in the United States suggested a biomarker-based solely biological definition of Alzheimer's disease. Even though this framework was designed for study, it has sparked discussion and problems when it comes to its use in clinical practice. Individuals with no cognitive impairment, for instance, despite having biomarker proof of both amyloid and tau pathology, seldom develop clinical symptoms over their lifetime. When AD pathology is present as comorbidity, biomarkers with an AD pattern can also be discovered in other brain diseases [6]. AD might be regarded as a purely biological illness with no clinical component or individual status based on ATN status. Though neuropathologists declared in 2012 that "there is an agreement to separate the clinicopathologic term 'AD' from neuropathologic alteration," by isolating AD from a clinical phenotype, the illness becomes identical with AD neuropathological adaptations [7]. As a result, the term Alzheimer's disease encompasses a spectrum of symptoms ranging from mild cognitive impairment to severe dementia. A model is a machine learning system that has been taught to recognize particular sorts of patterns via the use of a method. That is, it analyzes data and uncovers latent structures in a dataset [8]. The feature extraction and known replies of a dataset create the formula that relies on the input and output functions and applies it to new data to predict the response. As a consequence, the algorithm of the model takes a set of data for training and then produces a technique for predicting the output that can be preserved for later use [9]. In machine learning research, small datasets are frequently seen as a flaw. Small datasets, for instance, may constrain the applicability of the trained machine approach since they are less likely to capture all variation and ignore rare data points [10]. Furthermore, if the trained machine model is trained on a smaller database, it may get more data-dependent. When an ML model overfits the training dataset, this is referred to as data reliance. And the categorization was dependent on remembering the data or co-existing characteristics rather than on generic picture features. Disease categorization using medical pictures gathered from several sites, when each site utilizes various scanners with various image formats, is an excellent example. When certain research locations are strongly linked to a certain illness, excellent classification accuracy can be attained by differentiating picture format on the data collecting site rather than pathological features [10]. According to research, the situation may increase if individuals can diagnose AD early and begin treatment [11]. It is imperative that they anticipate the progression of the disease from a mild state to dementia in order to do so. The use of machine learning technologies can aid in the accurate prediction of early Alzheimer's disease, as it can in many other fields. Although there are many machine learning systems available, their predictions are unreliable and inaccurate [12]. It is also difficult for them to find the local and global optimal solution to overfitting and underfitting [13].

The study utilizes machine learning (ML) techniques to determine the severity of Alzheimer's disease. The dataset used in the study comprises MRI images with four different levels of severity. We used a hybrid of 12 feature extraction methods to diagnose Alzheimer's disease severity using MRI images. Six traditional machine learning methods are used to diagnose Alzheimer's disease. The techniques include decision tree (DT), K-nearest neighbor (KNN), Linear Discrimination Analysis (LDA), Naïve Bayes (NB), Support Vector Machine (SVM), and ensembled learning, including Bag, Adaboost, and RUSBoost methods. In the training process, optimization is done to find the best solution for each classifier. Moreover, a CNN model is taught to recognize patterns using a machine learning technique. The remainder of the paper is organized as follows: Section 2 presents the related work, while Section 3 describes the methods and materials used in the study. Section 4 presents the results of the experiments and a discussion of the findings. Finally, Section 5 provides the conclusion and future work.

## 2 Literature review

Deep learning (DL), a state-of-the-art ML technique, has outperformed standard ML algorithms in identifying complicated structures in high-dimensional data. Using multimodal neuroimaging data is a potential diagnostic classification technique for AD. Furthermore, when neuroimaging data is scarce, hybrid methods based on DL for feature extraction might improve AD classification [14]. Karaglani et al. used AutoML technology to evaluate a high-throughput dataset from AD blood tests to create reliable prediction approaches for application as diagnostic biosignatures. The findings suggested less invasive blood-based diagnostic diagnostics for AD, pending clinical confirmation depending on laboratory experiments. They also emphasized the importance of AutoML in the identification of biomarkers [15]. Naik et al. investigated the impact of multiple ML classifiers in MRI and the usage of SVM with various multimodal scans for identifying patients with AD/MCI against healthy controls. Findings were reached based on different classifier techniques and the presentation of the best multimodal paradigm for AD categorization [2]. Uysal and Ozturk used neuroimage analysis to detect dementia early in AD. They believe that the hippocampus's volumetric decrease is the most crucial indication of AD. ITK-SNAP, a semi-automatic segmentation software, was used to generate volume information, and a data set was constructed depending on age, gender, diagnosis, and right and left hippocampus volume values. ML algorithms were used to make the diagnosis based on hippocampus volume data. They find that brain MRIs may be utilized to identify individuals with AD, Mild Cognitive Impairment (MCI), and Cognitive Normal (CN) from one another, whereas most research could only differentiate AD from CN [16]. Gaudiuso et al. showed that using Laser-Induced Breakdown Spectroscopy (LIBS) and ML to examine micro-drops of plasma samples from AD patients and healthy controls resulted in accurate classification. After collecting LIBS spectra from 67 plasma samples from 31 Alzheimer's patients and 36 healthy controls (HC), the researchers examined the data. With an overall accuracy of 80%, a specificity of 75%, and a sensitivity of 85%, they accurately diagnosed late-onset AD (beyond 65 years old) [17]. Rzhikova et al. proposed a novel technique for diagnosing AD depending on CSF using near-infrared (NIR) Raman spectroscopy and ML analysis. Raman spectroscopy can examine a biological fluid's complete biochemical makeup at once. It offers much promise for detecting tiny changes unique to AD, even at the initial stages of the disease [18]. Artificial neural networks (ANNs) and support vector machine discriminant analysis (SVM-DA) statistical methods were employed to achieve 84% sensitivity and specificity for the discrimination of AD and HC patients. The Raman spectroscopic examination of CSF described in this study can enhance existing clinical testing, enabling rapid, accurate, and low-cost detection of AD in the early stages. While the results of this study were

promising with limited sample size, further technique validation on a larger scale will be required to determine the approach's true potential [18].

Tian et al. decided to look at the retina, specifically the retinal vasculature, instead of undertaking AD-related dementia assessments. All through the process, highly modular ML approaches were used. They also incorporated a saliency analysis to improve the interpretability of this process. According to the saliency analysis, tiny vessels in retinal pictures convey more data for detecting AD, consistent with previous research [10]. A traditional SVM and a deep CNN method were used by Bron et al. [19]. based on structural MRI images pre-processed into modulated gray matter (GM) maps with either a minimum or substantial amount of preprocessing. In the study, both deep and conventional classifiers performed similarly well for the classification of AD, and their performance only marginally deteriorated when used with an external cohort. By combining information-gain-based feature selection with an Ada-Boost classifier, Mienye et al. [20] highlighted the effectiveness of ML in early Chronic Kidney Disease (CKD) identification. Earlier, Mienye et al. [21] used an improved sparse autoencoder with Softmax regression to address unbalanced medical datasets, resulting in superior disease prediction outcomes for CKD, cervical cancer, and heart disease. While employing ML models in conjunction with SHapley Additive exPlanations (SHAP) for greater interpretability, Obaido et al. [22] improved hepatitis B diagnosis, underlining the value of certain characteristics like bilirubin in predicting outcomes.

Bari Antor et al. reported the results and analyses of several ML models for diagnosing dementia. The system was created using the Open Access Series of Imaging Studies (OASIS) dataset. The data was examined and used in a variety of ML models. SVM, logistic regression, decision trees, and random forests were utilized for prediction. The system was running without fine-tuning for the first time and then with fine-tuning for the second time. When the findings were compared, it was discovered that the support vector machine produced the best outcomes of the models [12].

Dogan et al. [23] developed a model of primate brain patterns based on EEG signals for the detection of AD. Using a directed graph to extract features from the primate brain's connectome, this method demonstrated high accuracy in identifying AD patients from healthy controls. In order to enhance the accuracy of AD detection, the model is able to generate a set of features from EEG signals. Using brain images, Kaplan et al. [24] developed a feed-forward Local Phase Quantization Network (LPQNet) for AD detection. Based on feature generation and selection through multilevel processing, their model demonstrated remarkable classification accuracy across several datasets. LPQNet stands out for its combination of high accuracy and low computational complexity, which makes it a valuable tool for diagnosing Alzheimer's disease. Kaplan et al. [25] developed the ExHiF model for the detection of AD using CT and MR images. They combined exemplar histogram-based features with neighborhood component analysis to achieve 100% classification accuracy. In terms of medical image classification for AD, this model is innovative in its feature extraction process inspired by vision transformers. Table 1 provides a detailed comparison of related works that have leveraged ML to diagnose AD.

## 3 Methods and materials

### 3.1 Feature extraction

The Gray Level Co-occurrence Matrix (GLCM) and related texture feature calculations are used in image analysis. The GLCM is a tally of how often different gray level combinations co-occur in an image or segment, resulting in a picture of pixels with different intensities (a specific gray level). In texture feature calculations, the components of the GLCM are utilized to

**Table 1. A comparison of the findings of the ML approaches used to diagnose AD.**

| Authors | Year | Aim | Disease | Method | Results |
|---|---|---|---|---|---|
| Ryzhikova et al. [18] | 2021 | Analysis of cerebrospinal fluid | AD | Raman spectroscopy, ANN, SVM-DA | CSF's described Raman spectroscopic analysis can supplement current clinical testing, allowing for quick, accurate, and low-cost early AD diagnosis. |
| Tian et al. [10] | 2021 | Analysis of retinal vasculature | AD | Highly modular machine learning | The saliency study revealed that tiny vessels in retinal pictures convey more information for detecting AD, consistent with previous research. |
| Bron et al. [19] | 2021 | MRI image processing | AD | SVM, CNN | Deep and conventional classifiers performed equally well in the categorization of AD, and their function only marginally deteriorated when deployed to the external population |
| Chang et al. [26] | 2021 | Imaging and cerebrospinal fluid (CSF) levels of amyloid-$\beta$1-42 (A$\beta$42), total tau protein, and hyperphosphorylated tau (p-tau) | AD | SVM, logistic regression, random forest, and naïve Bayes | The findings showed that ML combined with new biomarkers and numerous factors might improve the sensitivity and specificity of AD diagnosis. HPLC for biomarkers and ML algorithms may aid clinicians in identifying Alzheimer's disease in outpatient clinics |
| Rodriguez et al. [27] | 2021 | Identifying repurposing drug | AD | logistic regression, SVM, boosted random forest models, and two-layer fully connected neural networks | The DRIAD technique may be utilized to identify medicines that might be quickly assessed in a clinical trial following additional verification and identifying of relevant pharmacodynamic biomarker(s). |
| Ficiarà et al. [28] | 2021 | Analysis of cerebrospinal fluid | AD | ANOVA, Kruskal-Wallis test, Post-hoc tests, Linear SVM, LR model | The findings indicate the role of iron dysregulation in the etiology and development of dementia, as well as its possible interaction with biomarkers (Tau protein and Amyloid-beta) |
| Karaglani et al. [15] | 2020 | Analysis of Blood-Based Diagnostic Biosignatures | AD | AutoML, Ridge Logistic Regression, Support Vector Machines, Random Forests | These findings suggested less invasive blood-based diagnostic diagnostics for AD, which were still awaiting clinical confirmation obtained from laboratory assays |
| Naik et al. [2] | 2020 | Multimodal diagnostic classification | AD | KNN, SVM | Multimodal methods provide more important information concealed when a single modality is investigated, and they can reveal unique and distinct data characteristics by identifying multiple correlations between the data. |
| Gaudiuso et al. [17] | 2020 | Laser-Induced Breakdown Spectroscopy | AD | Quadratic Discriminant Analysis, linear discriminant analysis | These findings were acquired by selecting features from the difference spectra manually (i.e. all the features that appeared as positive or negative peaks in difference spectra) |
| Nanni et al. [29] | 2020 | MRI image processing | AD | Transfer Learning, 3D-CNN | These findings offer up new possibilities for using transfer learning in conjunction with neuroimages for the automated early identification and prognosis of Alzheimer's disease, even if the system is pre-trained on generic images |

produce a measure of intensity variation (a.k.a. image texture) at the pixel of interest [30]. For obtaining texture features, the LBP is a helpful tool. Face detection and pattern recognition algorithms typically use this strategy. The LBP operator converts an image into an array or an image of integer labels that describe how the picture looks on a small scale. The operator may be tweaked to work with different community sizes. Any neighborhood radius and the number of pixels may be achieved by employing circular neighborhoods and bilinearly interpolating the pixel values [31]. Furthermore, rather than utilizing angular neighbor points, the primary technique of RLBP is to determine the mean of points across each radial (over a circle). Local ternary patterns (LTP), unlike LBP, do not utilize a threshold constant to divide pixels into 0 and 1 but instead use a threshold constant to divide pixels into three values [32]. Statistical

geometrical metrics include area, centroid, convex area, convex hull, equivalent diameter, Euler number, extent, extrema, bounding box, filled size, major axis length, eccentricity, minor axis length, orientation, perimeter, and solidity. The boundary intersection-based signature (BIBS) examines the contents of a shape's border, particularly those with concave contours. The design of this technique relies heavily on the intersection locations of any linked line from the form center to its outline [33]. A shape signature is a one-dimensional representation of a shape border and cannot describe open borders with many intersecting locations. As a result, BIBS examines the boundary contents of forms, particularly those with concave contours [34]. The PCA filter feature is the mean of the input image's PCA coefficient. Independent component analysis (ICA) is a method for obtaining additive subcomponents from a multivariate signal using a computer program. This is done by assuming that the subcomponents are statistically independent non-Gaussian signals. Blind source separation is a particular instance of ICA [35]. The list of the feature extraction approaches is depicted in Fig 1.

A series of Gabor filters with varying frequencies and orientations may be beneficial for extracting relevant information from a picture. Two-dimensional Gabor filters are defined as follows in the discrete domain:

$$G_c(i,j] = Bexp\left(-\frac{i^2 + j^2}{2\sigma}\right)$$
$$cos(2\pi f(icos\theta + jsin\theta))$$

(1)

$$G_s(i,j] = Cexp\left(-\frac{i^2 + j^2}{2\sigma}\right)$$
$$sin(2\pi f(icos\theta + jsin\theta))$$

(2)

where B and C are to be identified, normalizing factors 2-D Gabor filters have several uses in image processing, mainly feature extraction for texture analysis and segmentation. The frequency searched for in the texture is defined by f. We may seek texture oriented in a specific direction by changing. We can alter the size of the picture region being examined or the support of the basis by changing [36]. The following technique calculates long energy using a discrete wavelet spectrum, shown in Table 2. The logarithm of the absolute Shift zero-frequency component of the discrete Fourier transform to the center of the spectrum is LE.

The Model-Based Feature returns the Hausdorff fractal dimension of an item described by a binary picture. An item has nonzero pixels, while the backdrop has zero pixels. Ultimately, the width and heights of the binary sub-border of the images are returned by Conventional Shape Signature.

## 3.2 Feature reduction

In ML, it is commonly assumed that the more characteristics we have, the better our prediction. However, this is not always the case. If we keep raising the number of features, the efficiency of our ML algorithm will eventually deteriorate. If we maintain the number of training samples constant while expanding the number of dimensions, the predictive power of our ML model rises at first but subsequently begins to decline. As the number of dimensions grows, each feature may take on a wider range of values; therefore, as the dimensions grow, we must raise the number of data samples to guarantee that there are numerous samples with each value combination. The number of samples required for successful prediction often grows logarithmically as the dimensions increase [37]. The Principal Component Analysis (PCA) is a technique for reducing dimensionality through the transformation of correlated features in

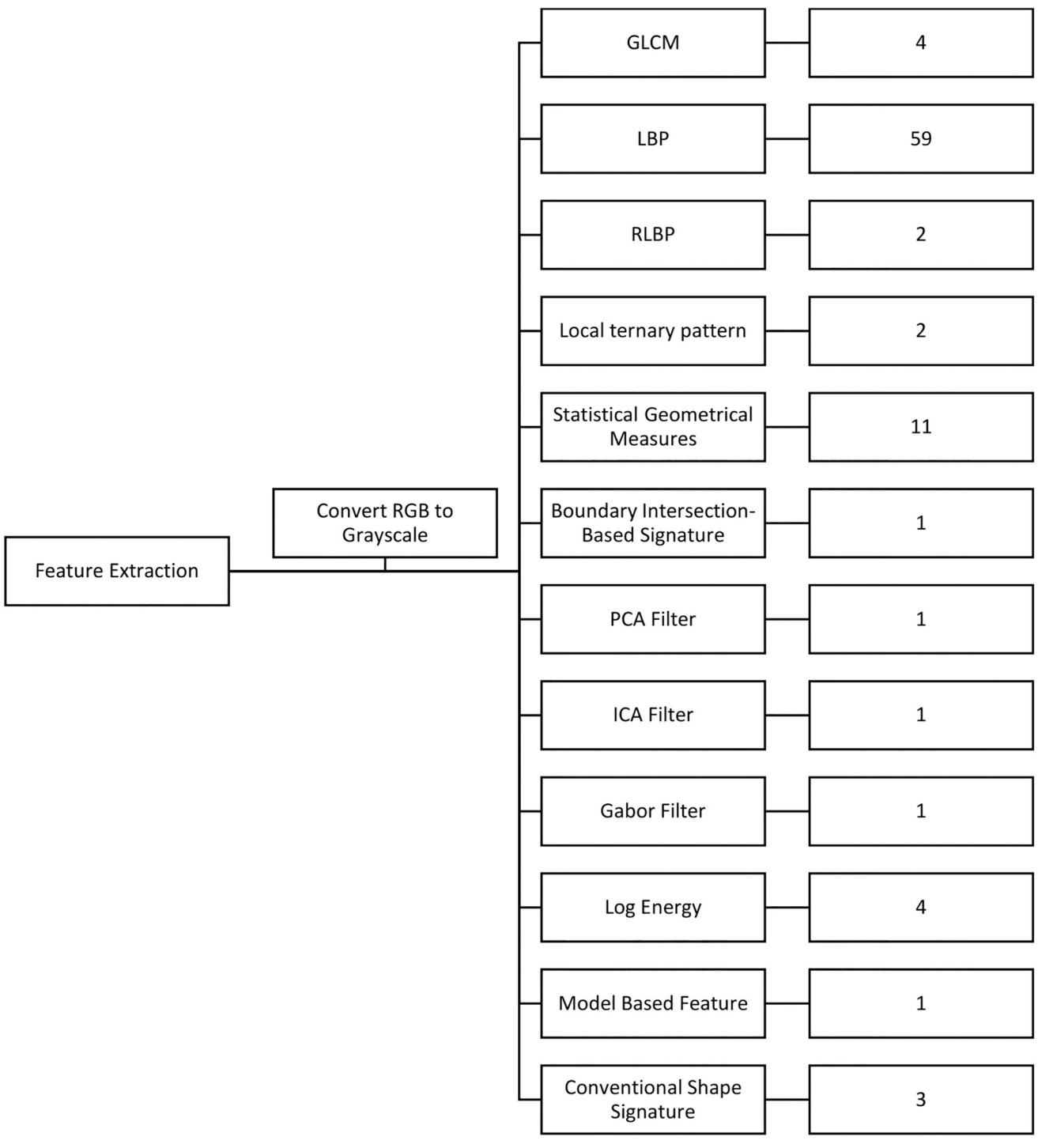

**Fig 1. The diagram of all feature extraction methods.**

high-dimensional spaces into uncorrelated features in low-dimensional spaces. The term "principal components" refers to these uncorrelated features. PCA is an orthogonal linear transformation, meaning all principal components are perpendicular to another one. It modifies the data so that the first component seeks to explain as much variation as possible from the

**Table 2. Logic-energy pseudo-code.**

| Algorithm: Log-Energy (LE) |
|---|
| F: features; Im: input image; level: level of wavelet transformation |
| **for** i = 1:level [a1, a2, a3, a4] = DWT(Im); Discrete wavelet transformation subbands **do** |
| A(i) = mean (log(abs(fftshift(fft2(a1))))); |
| H(i) = mean (log(abs(fftshift(fft2(a2))))); |
| V(i) = mean (log(abs(fftshift(fft2(a3))))); |
| D(i) = mean (log(abs(fftshift(fft2(a4))))); |
| **end for** |
| RA = sum(diff((A(:)))); |
| RH = sum(diff((H(:)))); |
| RV = sum(diff((V(:)))); |
| RD = sum(diff((D(:)))); |
| F = [RA, RH, RV, RD]; |

original data. It's an unsupervised algorithm that doesn't take the class labels into account. When we apply PCA to reduce dimensionality, we create a d×k–dimensional transformation matrix. This matrix may be used to map a sample vector x onto a new k–dimensional feature space with fewer dimensions than the original d–dimensional feature space [38]:

$$x = (x_1, \ x_2, \ \ldots, \ x_d],$$
$$x \in R^d \rightarrow z = (z_1, \ z_2, \ \ldots, \ z_d], \tag{3}$$
$$z \in R^k, \quad d" k$$

The original d-dimensional data was changed into this new k-dimensional subspace due to this transformation (typically k"d). The most variation will be seen in the first primary component since consecutive principal components must be uncorrelated (orthogonal) to the primary principal components, even if the input characteristics are connected. And the most considerable variance will be found in all subsequent significant features. The key factors that emerge will be unrelated (mutually orthogonal) [39].

## 3.3 Machine learning (ML)

ML is the study of computer systems that learn through inference and patterns through algorithms and statistical models without being explicitly programmed. In recent years, machine learning algorithms have been developed independently for a variety of purposes, including health, finance, and agriculture [40]. In supervised machine learning, SVMs are used to apply classification techniques to problems involving two groups [41]. SVMs are reliable and fast classification techniques that are excellent for handling sparse data. SVMs are a group of supervised learning algorithms that are used to solve regression and classification problems. The decision tree approach is used to categorize data in ML systems. Based on the training data, the goal of a decision tree is to construct the smallest tree possible [42].

A split test is conducted in the decision tree's core node and a target class example is expected in the leaf node of this supervised classification method. KNN is a nonparametric feature similarity-based algorithm. It is a successful algorithm for pattern recognition. Data points are classified according to their closest neighbors using a straightforward classifier. KNN is likely to be a good fit for studies involving large databases. As a result of the enormous amount

of data available in medical databases, KNN is capable of accurately predicting a new sample point class. KNN classification algorithms based on dimensionality reduction outperform existing probabilistic neural network schemes in terms of average accuracy, specificity, sensitivity, recall, precision, Jaccard and Dice coefficients, reduced data dimensionality, and computational complexity, according to the research results [43].

### 3.4 Convolutional Neural Network (CNN)

As discussed in the chapter before, CNNs consist of neurons with trainable weights and biases. In each neuron, some inputs are received, a dot product is calculated, and a non-linearity is applied if necessary [44]. The complete network delivers a single differentiable score that ranges from raw image pixels on one end to class scores on the other. All of the methods and techniques we described for conventional neural networks still apply to the last (completely connected) layer, which still has a loss function (such as SVM/Softmax). The CNN is particularly adept at identifying objects, people, and scenery in images by searching for patterns in them. Medical image processing has increasingly used CNNs to detect malignancies in the breast, brain, and teeth [45]. As well as classifying image data, these algorithms are also useful for classifying other types of data, such as audio, time series, and signals [45]. Advanced machine learning and optimization approaches have demonstrated intriguing uses in recent medical research. As an illustration, consider an ensemble strategy that uses LSTM neural networks and hybrid data resampling to optimize fraud detection [46] or medical imaging that uses genetic algorithms to improve picture accuracy before utilizing DenseNet for classifications [47].

### 3.5 Classification performance metrics

The confusion matrix is a specific tool that accurately measures categorization performance. Learning a few definitions are required to understand the confusion matrix [48]. But before we get into the theories, let's look at a fundamental confusion matrix for binary or binomial classification with two classes (say, Y or N). The potential of a classifier to select all of the examples that must be chosen is referred to as sensitivity. A perfect classifier will determine all true Ys and exclude any true Ys. There will be no false negatives to express it in a different way. In the absence of a classifier, true Ys will be missed, resulting in false negatives. A classifier is defined as being accurate if it is able to select all instances that need to be picked and to reject all instances that need to be denied [49]. Fig 2 illustrates the workflow and conceptual diagram in our proposed approach.

## 4 Results and discussion

### 4.1 Data collection

There are 426 individuals in the clinical medical dataset, comprising 1229 records of prospective patients. The pictures MRI segmentation AD dataset contains four classes of images in both the training and testing sets, totaling about 5000 images, each separated by Alzheimer's severity, indicated in Fig 3:

i. Moderate Demented

ii. Mild Demented

iii. Very Mild Demented

iv. Non-Demented

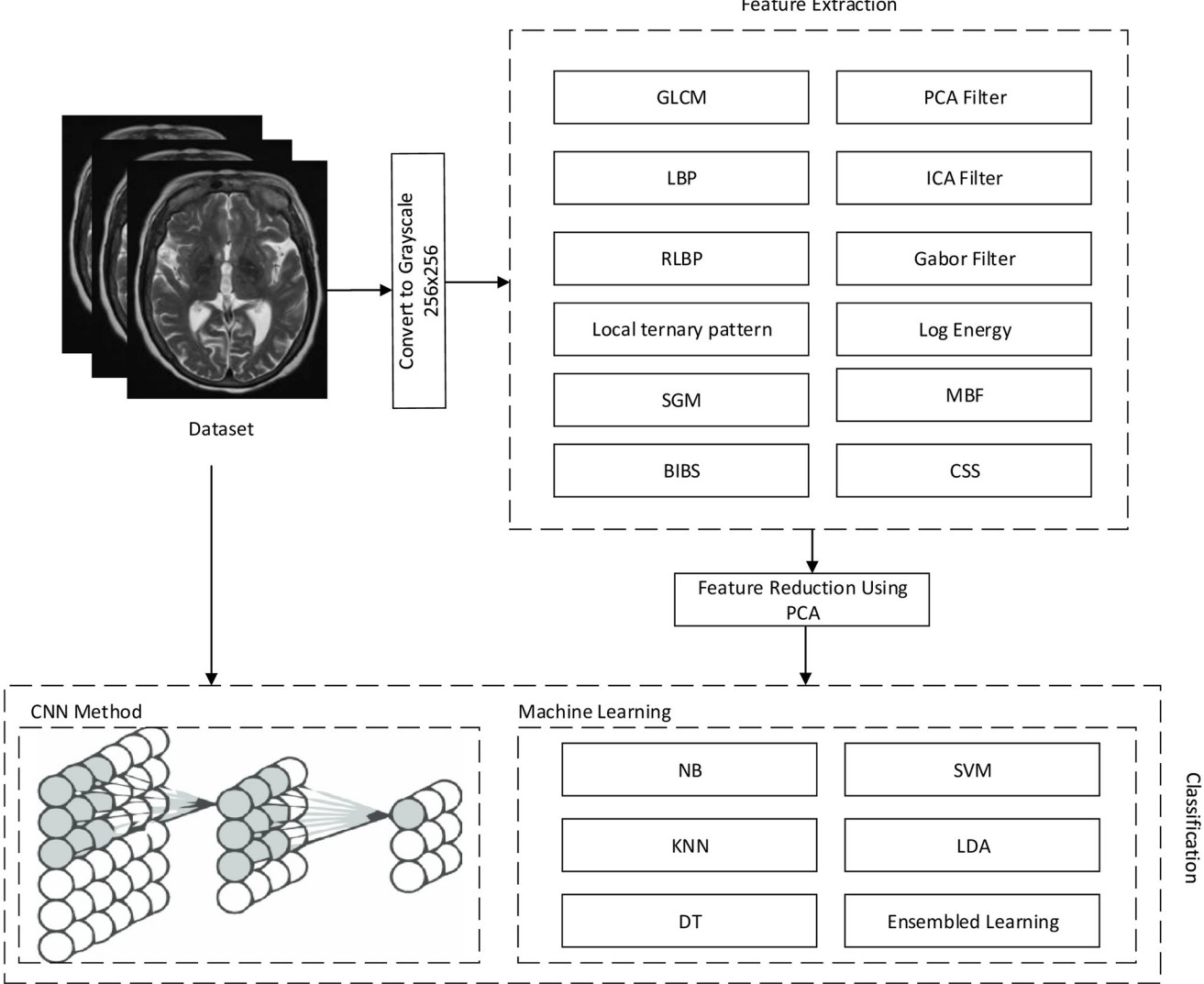

**Fig 2. Conceptual diagram of the proposed method.**

### 4.2 Pre-processing results

This paper used a hybrid of 12 feature extraction methods to diagnose the AD severity using MRI images. These feature extraction methods consisted of GLCM, LBP, RLBP, LTP, SGM, BIBIS, PCA Filter, ICA Filter, Gabor Filter, Log-Energy, Model-based Feature, and conventional shape signature, as demonstrated in Fig 1. The total number of features is 90 image features. After rescaling and transforming the image to double and grayscale images and normalization, the feature extraction is done. To reduce the computation time and optimize the training process's computation, the PCA feature reduction method was used. Fig 4 shows the scree plot and normalized cumulative sum of eigenvalues (NCSE). Based on the results, 33 first eigenvalues illustrate 100 of all feature effects of the classification. Furthermore, the output labels are divided into four categories: 1) Non-Demented, 2) Very Mild Demented, 3) Mild Demented, and 4) Moderately Demented.

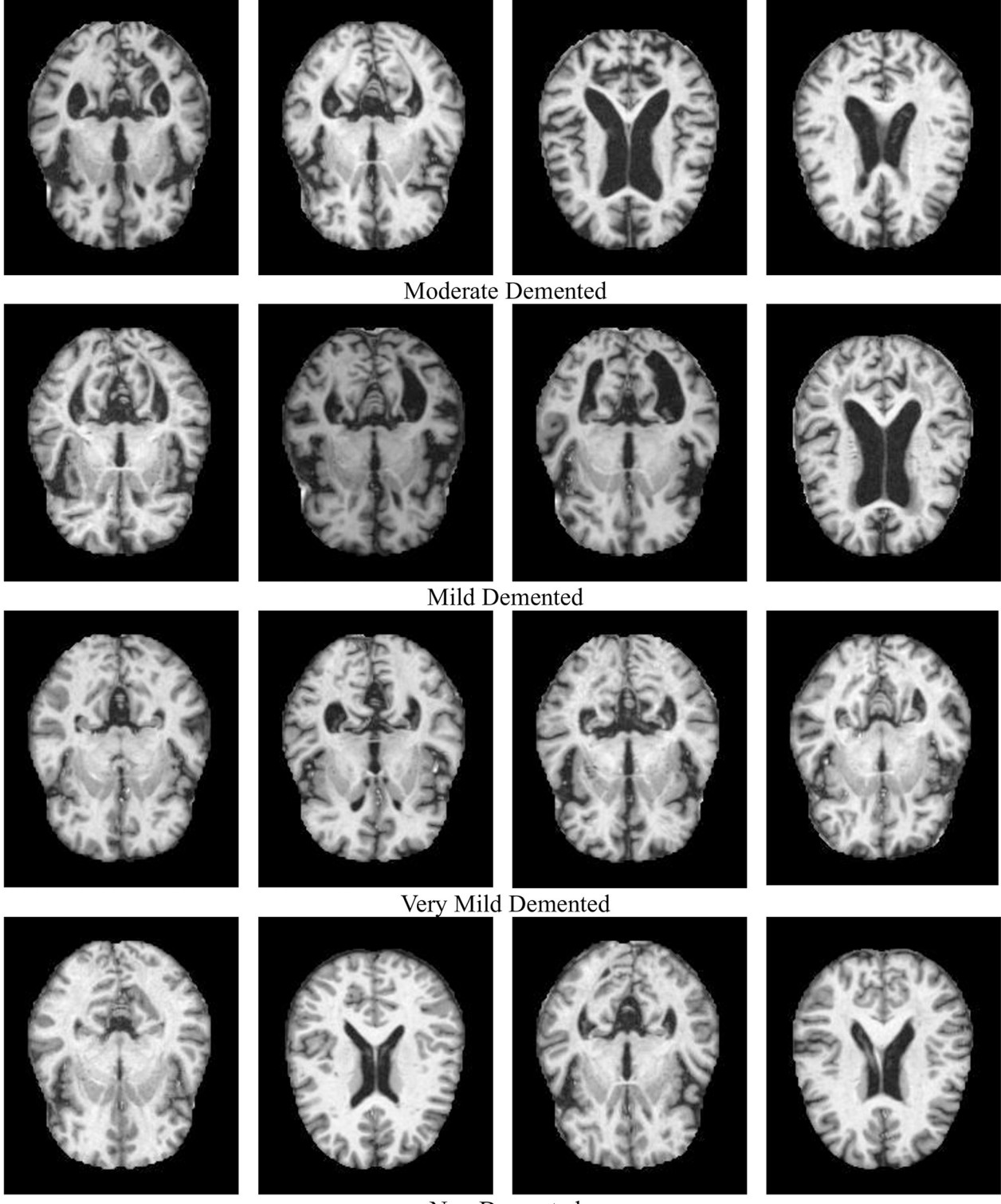

**Fig 3. Dataset outcome of different dementia stages.**

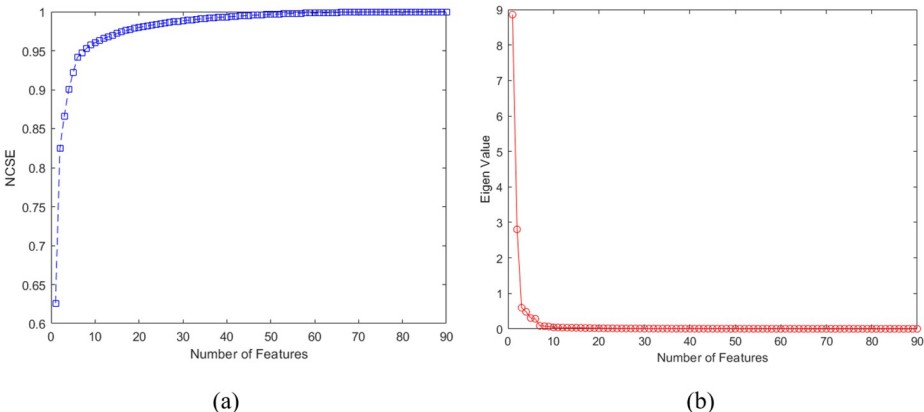

(a) (b)

**Fig 4. Results of PCA for feature reduction.** The first 33 features include 100% variance of all 99 features.

## 4.3 Classification results

In this work, six traditional ML methods were employed to diagnose AD. The techniques include DT, KNN, LDA, NB, SVM, and ensembled learning includes Bag, Adaboost, and RUS-Boost methods. In the training process, optimization is carried out to find the best solution for each classifier. DT is the first classifier used in the training process. Based on the optimization results in Fig 5 (middle column).

Regarding the results, 138 slits of the DT method with Towing rule reached minimum error in the training process. The classification confusion matrix is illustrated in Fig 5 (left column). The total number of images for classification is 4000 MRI images, such that each class includes 1000 images. The training process performed used 33 reduced features with four labels. Based on the confusion matrix, the diameter of the matrix illustrates the actual values or number of images that were diagnosed correctly in each class.

Based on the results from normal images (1- Non-Demented), 589 images were diagnosed correctly. It includes 58.9% of non-demented images. Also, 23.8%, 15.9%, and 1.4% are misdiagnosed on 2-Very Mild Demented, 3- Mild Demented, and 4- Moderate Demented, respectively. Moreover, between 1000 Very Mild images, 530(53%) were diagnosed correctly. The sensitivity of the DT method for the diagnosis of AD is 58.9%, 53%, 52.1%, and 85.4% for the diagnosis of classes 1, 2, 3, and 4. To better illustrate the classification results, the false-positive rate versus the true positive rate is the ROC curve represented in Fig 5 (right column). The

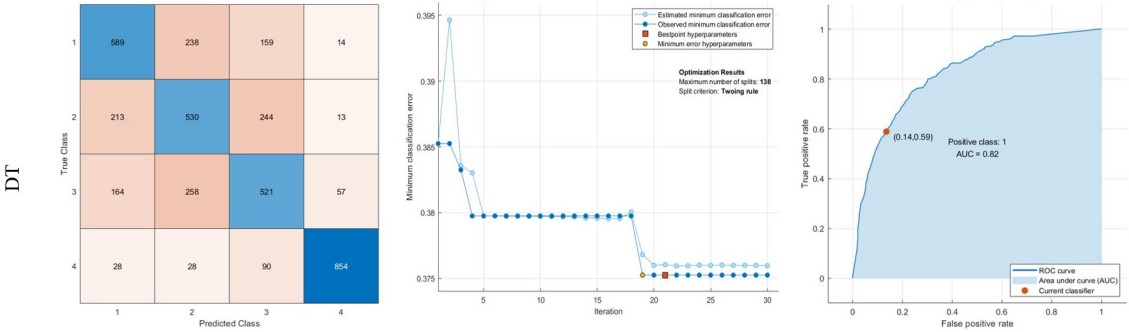

**Fig 5. Results of classification using DT: Confusion matrix, middle: Optimization results, Right: ROC curve.**

positive class is determined based on normal images. If the curve tends to left-up or minimum false positive rate and maximum true positive rate, it would be desirable for AD classification. Therefore, if the area under the curve (AUC) is high, it is better than other classifiers. The AUC value for the DT method is 82%.

The presented optimized ensemble learning (En) includes three classifiers, AdaBoost, Bag, and RUSBoost. The learner type is the DT method, and 288 learners are engaged to classify the features. The learning rate is also 0.001. The maximum number of splits is 2221. These hyper-parameters are optimized to decrease the maximum classification error. Based on the confusion matrix of the En method, it can diagnose 87.7% of Moderate Demented images. The AUC value of the En method is also 93%. The results are reported in Fig 6

Moreover, the KNN method is optimized with k = 12 neighbors with correlation distance metrics. It results in minimum classification error. This classifier's maximum sensitivity belongs to the fourth class with 90.0% sensitivity. 6.6%, 2.3%, and 1.1% of images are misdiagnosed in this class to 3,2 and 1. The outcomes as demonstrated in Fig 7.

The LDA method is also prosperous in diagnosing 85.7% of 4 (Moderate Alzheimer). Moreover, in the NB method, the sensitivity of classes 1, 2, 3, and 4 are 70.5%, 49.9%, 61.1%, and 81.3%, respectively. It is trained with a Gaussian kernel to minimize classification error. Finally, SVM is trained with a linear kernel. The AUC of SVM is also 89%. The results for LDA, NB, and SVM are indicated in Figs 8–10, respectively.

In this section of the study, a CNN design is introduced for determining the severity of Alzheimer's disease It includes 20 layers consisting of four convolutional layers, four

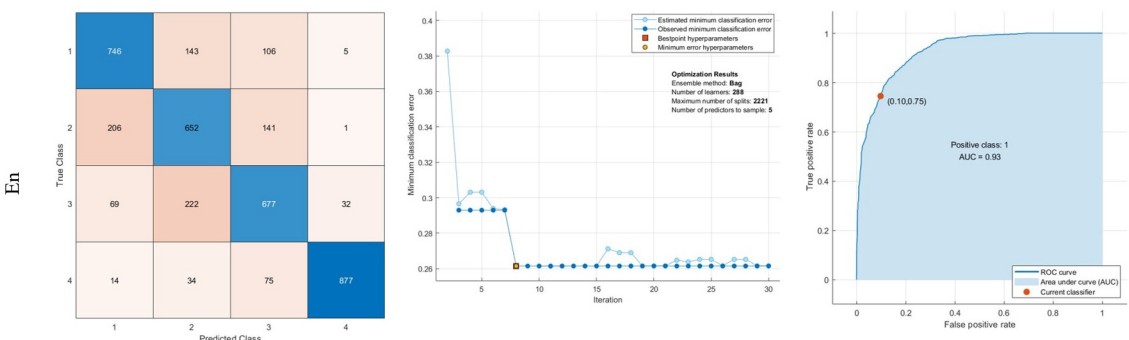

**Fig 6. Results of classification using En: Confusion matrix, middle: Optimization results, Right: ROC curve.**

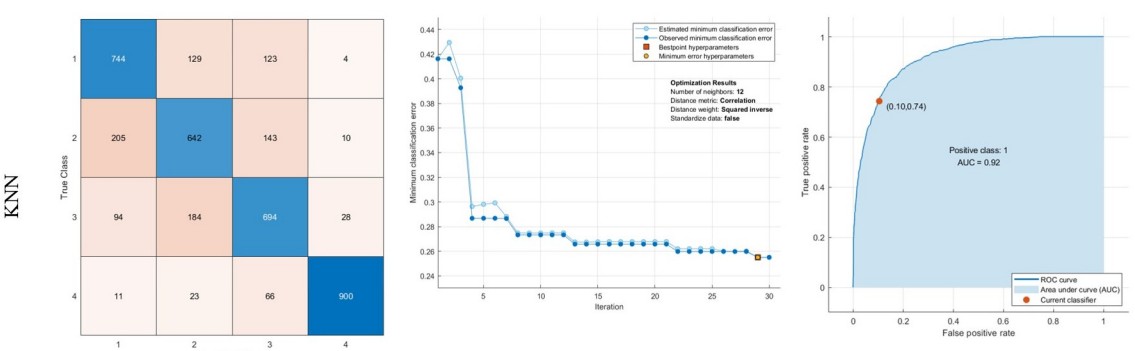

**Fig 7. Results of classification using KNN: Confusion matrix, middle: Optimization results, Right: ROC curve.**

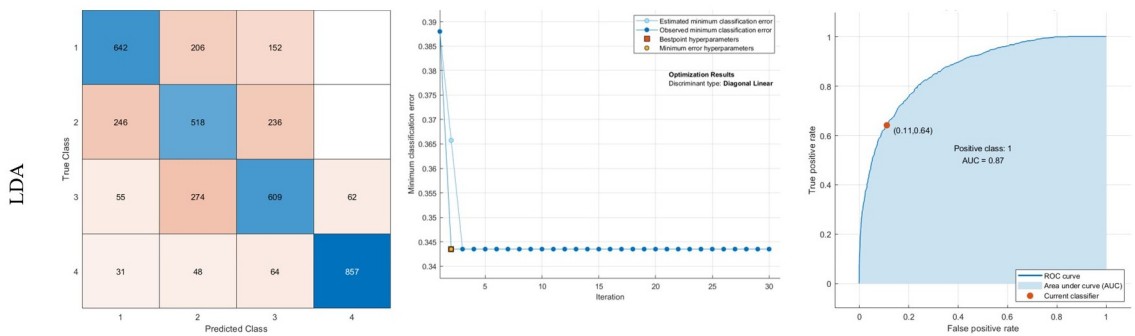

**Fig 8. Results of classification using LDA: Confusion matrix, middle: Optimization results, Right: ROC curve.**

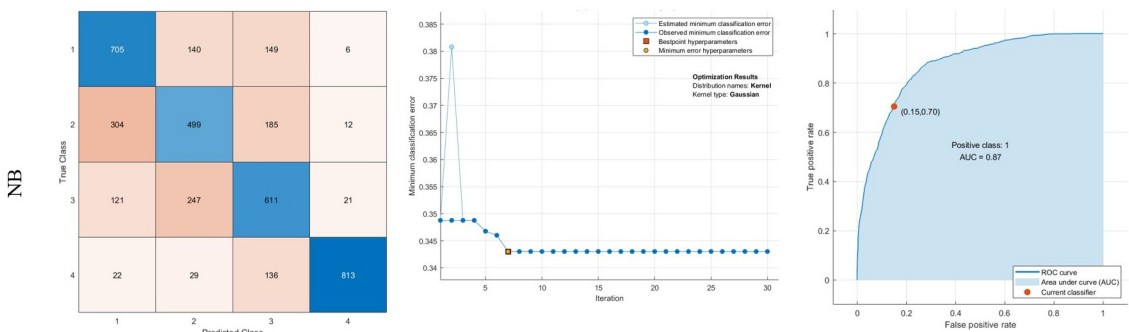

**Fig 9. Results of classification using NB: Confusion matrix, middle: Optimization results, Right: ROC curve.**

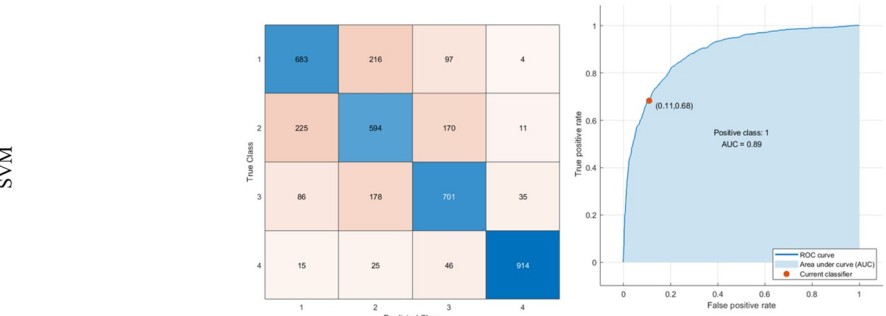

**Fig 10. Results of classification using NB: Confusion matrix, Right: ROC curve.**

normalization layers, four activation layers, and 3 Maxpooling layers. It begins with the image dataset input layer and is evaluated with a 4-label output layer. Two Softmax and fully connected layers are also necessary for classification architecture, and a 50% dropout layer is added to increase the accuracy. Fig 11 illustrates the architecture of the proposed CNN method.

The dataset consists of 4000 PNG images of brain MRI results classified into four categories: Non-Demented (Normal), Very Mild Demented, Mild Demented, and Moderately Demented.

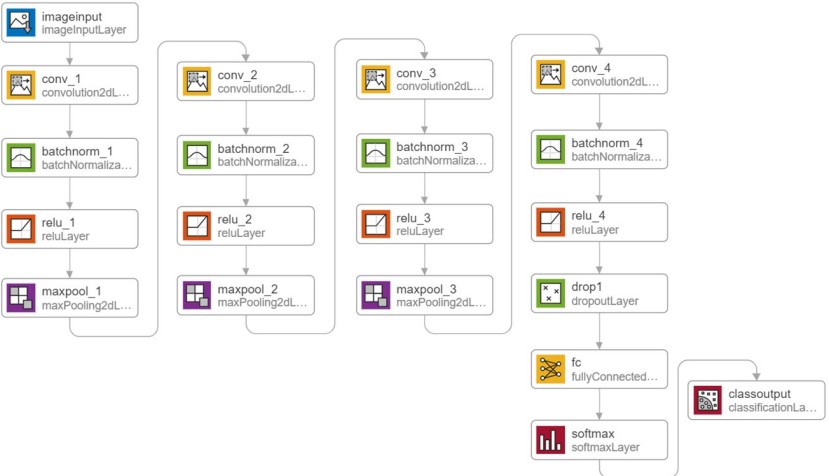

**Fig 11. An overview of the architecture of the CNN method presented in this paper.**

70% of all data is used for training in the training process, and 30% is allocated for testing the output network. The training process is done with 13 epochs and 3500 iterations with a gradient descent algorithm. The accuracy and loss of the training process are shown in Fig 12.

The classification results are offered in a confusion matrix, as represented in Fig 13. The green cell (diameter of the matrix) illustrates the true values, and the red cells are false results. To test the network's fitness, the results of the confusion matrix of test samples can be seen in Fig 13(a). Based on the results of the classification, the training process results in 100% accuracy for the classification of input images, as shown in Fig 13(b). From 300 mild images, 288 (96.0%) of them are diagnosed correctly. In other words, 4 (1.2%) of the mild class is misdiagnosed in the normal category and 8 (2.8%) is located in the very mild subset. Therefore, the sensitivity of the CNN for diagnosing mild AD is 96%. The presented CNN method diagnosed moderate AD accurately with 100 sensitivity. Moreover, 95.3% of normal images are

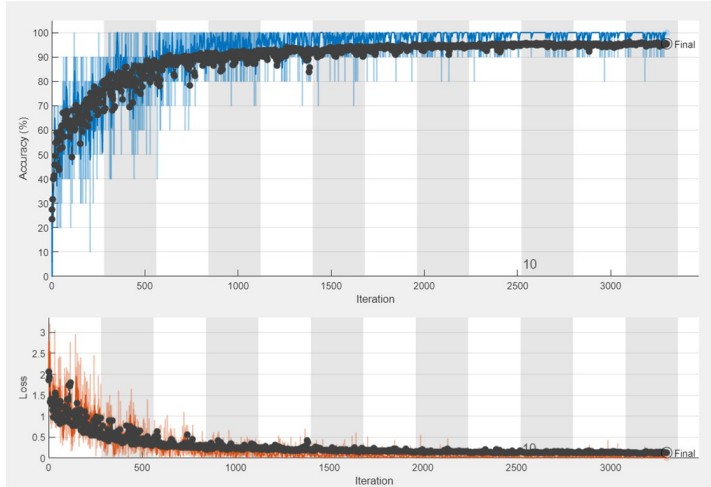

**Fig 12. The training process in the proposed CNN model.**

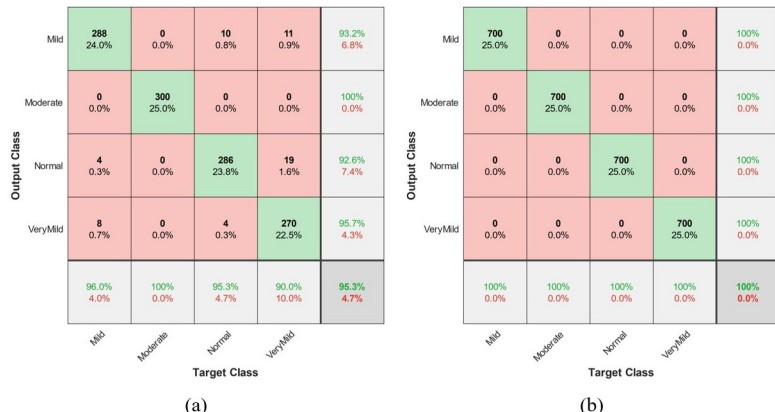

(a)                                                          (b)

**Fig 13. The confusion matrix of the presented CNN method.**

recognized. The similarity of the very mild and normal images causes small errors in the networks. Hence, (4) 1.2% of normal images are allocated in a very mild class, and (19) 6.4% of very mild images are misdiagnosed in normal classes. Moreover, the mild and very mild images are sometimes similar, so that, 11 (3.6%) of the very mild image are in the mild class and 8 (2.8%) of the very mild image are in the mild class.

Consequently, the sensitivity of the very mild images is 90%. The precision of the network is the other criterion of classification shown in the vertical gray cells. For instance, the accuracy of the CNN network for diagnosing normal images is 92.6%. It means that 92.6% of all analyzed images as normal are normal. In other words, 19 very mild images and 4 mild images are allocated in the normal classes, decreasing the network's precision. As a result, the precision of the network for diagnosis of mild, moderate, normal, and very mild class are 93.2%, 100%, 92.6%, and 95.7%, respectively. Finally, the accuracy of the CNN network is presented in the right lower corner of the confusion matrix in Fig 13(a). The accuracy is the rate of all true diagnosed images. For our presented CNN architecture, the accuracy is 95.3%. The discrepancy of accuracy value between test and training samples is very low; therefore, the presented network lacks the overfitting problem.

To compare the presented CNN method with other machine learning approaches, the AUC values and accuracy of the methods are presented in Table 3. Based on the results, the accuracy and the AUC value for the presented CNN methods are higher than other machine learning classifiers. Moreover, the KNN method with 74.5% accuracy and 92% AUC is the second classification priority. Furthermore, the ensembled learning of Bag, Adaboost, and RUSBoost

**Table 3. The comparison of the machine learning methods.**

| Method | Accuracy | AUC |
|---|---|---|
| Naïve Bayes | 67.5% | 87% |
| SVM | 72.3% | 89% |
| KNN | 74.5% | 92% |
| LDA | 65.6% | 87% |
| DT | 62.4% | 82% |
| Ensemble Learning | 73.8% | 93% |
| **Presented CNN** | **95.3%** | **99%** |

methods raises the accuracy to 73.8% and 93% AUC. To conclude. It can be seen that the presented CNN method has higher accuracy in diagnosing Alzheimer's severity. Diagnosis of Alzheimer's patients from normal people is simpler than finding the severity of the disease. This is because of the close similarity between each class image. Therefore, the traditional machine learning methods with many feature extraction methods could not reach higher accuracy. However, the presented CNN solved this problem with 95.3% and 100% testing and training accuracy.

## 5 Conclusion

We utilized a hybrid of 12 feature extraction methods to diagnose the severity of AD using MRI data. In this article, GLCM, LBP, RLBP, LTP, SGM, BIBIS, PCA Filter, ICA Filter, Gabor Filter, Log-Energy, Model-based Feature, and conventional shape signature were employed as the feature extraction methods. The PCA feature reduction approach was utilized to minimize calculation time and optimize the training process computation. For diagnosing AD, six standard ML approaches were used, DT, KNN, LDA, NB, SVM, and ensemble learning methods such as Bag, Adaboost, and RUSBoost. Optimization was carried out during the training phase to identify the optimum solution for each classifier. In terms of the findings, 138 slits of the DT technique using the Towing rule achieved the lowest possible error throughout the training phase. According to the results, 589 pictures from normal images (1- non-demented) were accurately diagnosed. Non-demented pictures account for 58.9% of the total. In addition, 23.8%, 15.9%, and 1.4% of people with 2-Very Mild Demented, 3- Mild Demented, and 4- Moderate Demented, respectively, are misdiagnosed. Furthermore, 530 (53%) of 1000 Very Mild images were identified correctly. For diagnosis of classes 1, 2, 3, and 4, the DT technique has a sensitivity of 58.9%, 53%, 52.1%, and 85.4%, respectively. Three AdaBoost, Bag, and RUSBoost classifiers are included in the given optimal ensembled learning (En). The En approach could diagnose 87.7% of Moderate Demented pictures based on the confusion matrix with an AUC value of about 93%. Furthermore, the KNN technique was optimized using correlation distance metrics with k = 12 neighbors. The categorization inaccuracy was kept to a minimum. The most excellent sensitivity in this classifier belongs to the fourth class, which has a 90.0% sensitivity. 6.6%, 2.3%, and 1.1% of images in this class are diagnosed mistakenly as 3,2, and 1. The LDA technique also effectively diagnoses 85.7% of class 4 cases (Moderate Alzheimer's). Furthermore, the sensitivity of classes 1, 2, 3, and 4 in the NB approach was 70.5%, 49.9%, 61.1%, and 81.3%, respectively. To reduce classification error, it was trained with a Gaussian kernel. Lastly, a linear kernel was used to train the SVM. with an AUC of 89%.

The training procedure achieved 100% accuracy for the categorization of input images based on the findings of the proposed CNN technique. Nevertheless, out of 300 mild images in the test samples, 288 (96.0%) were properly diagnosed. As a result, CNN's sensitivity for detecting mild AD is 96%. The proposed CNN technique correctly diagnosed mild AD with 100 sensitivity. Furthermore, normal images are identified in 95.3% of cases. Consequently, the network's precision for mild, moderate, normal, and very mild class diagnosis is 93.2%, 100%, 92.6%, and 95.7%, respectively. The accuracy of our provided CNN architecture was 95.3%. According to the findings, the accuracy and AUC value for the provided CNN techniques were greater than other ML classifiers. Furthermore, the KNN technique was ranked second in classification accuracy, with 74.5% accuracy and 92% AUC. Besides, combining the Bag, Adaboost, and RUSBoost algorithms improved accuracy to 73.8% and 93% AUC, respectively. To sum it up, the proposed CNN technique was more accurate in diagnosing Alzheimer's severity.

## Author Contributions

**Conceptualization:** Danial Javaheri, Kasra Danesh, Junbeom Hur.

**Formal analysis:** Mohsen Ahmadi, Danial Javaheri, Junbeom Hur.

**Funding acquisition:** Junbeom Hur.

**Investigation:** Mohsen Ahmadi, Matin Khajavi, Kasra Danesh.

**Supervision:** Danial Javaheri, Junbeom Hur.

**Validation:** Danial Javaheri, Matin Khajavi, Kasra Danesh, Junbeom Hur.

**Visualization:** Danial Javaheri, Junbeom Hur.

**Writing – original draft:** Mohsen Ahmadi.

**Writing – review & editing:** Danial Javaheri, Matin Khajavi, Kasra Danesh, Junbeom Hur.

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
