## [Decision Letter · Decision Letter 0]

17 Jul 2023

PONE-D-23-12992Proposing a Deeply Supervised Adaptable Deep Convolution Neural Network for Diagnosisand Automated Categorization of Alzheimer’s Severity Stage Based on Multitask Feature Extraction ModelsPLOS ONE

Dear Dr. Javaheri,

Thank you for submitting your manuscript to PLOS ONE. After careful consideration, we feel that it has merit but does not fully meet PLOS ONE’s publication criteria as it currently stands. Therefore, we invite you to submit a revised version of the manuscript that addresses the points raised during the review process.

We look forward to receiving your revised manuscript.

Kind regards,

Jose Gerardo Tamez-Peña, PhD

Academic Editor

PLOS ONE

“This research was supported by Basic Research Program

through the National Research Foundation of Korea (NRF) funded by the

Ministry of Science and ICT (RS-2022-00166712).”

“YES - This research was supported by Basic Research Program through the National Research Foundation of Korea (NRF) funded by the Ministry of Science and ICT (RS-2022-00166712”

Reviewers' comments:

Reviewer's Responses to Questions

**Comments to the Author**

1. Is the manuscript technically sound, and do the data support the conclusions?

Reviewer #1: Partly

Reviewer #2: Yes

2. Has the statistical analysis been performed appropriately and rigorously? 

Reviewer #1: Yes

Reviewer #2: Yes

3. Have the authors made all data underlying the findings in their manuscript fully available?

Reviewer #1: No

Reviewer #2: Yes

4. Is the manuscript presented in an intelligible fashion and written in standard English?

Reviewer #1: Yes

Reviewer #2: Yes

5. Review Comments to the Author

Reviewer #1: Review Recommendation:

General identifications: This research explores the application of machine learning (ML) techniques, particularly deep learning, in the early diagnosis and severity classification of Alzheimer's disease using MRI images.

Alzheimer's disease, the most prevalent form of dementia, poses significant challenges due to its gradual progression and varying severity levels. However, recent advancements in neuroimaging techniques have paved the way for large-scale multimodal neuroimaging data, sparking increased interest in leveraging deep learning for automated Alzheimer's disease classification.

The study employs a hybrid approach that combines 12 feature extraction methods to diagnose the severity levels of Alzheimer's disease. In addition, six traditional machine learning algorithms are applied, including decision tree, K-nearest neighbor, linear discrimination analysis, Naïve Bayes, support vector machine, and ensemble learning methods. Each classifier is optimized during training to obtain the best solution.

Notably, the researchers also train a convolutional neural network (CNN) model using a machine learning system algorithm to identify specific patterns in the MRI images. The CNN architecture achieves an impressive accuracy of 95.3%, surpassing the performance of other traditional machine learning methods employed in the study.

Recommendations

1. Dataset Size and Diversity: Use of ADNI and Compare with 4 class of image.

2. This diversity enhances the generalizability of the findings and ensures that the models are not biased towards a specific population.

3. Clinical Validation: While accuracy rates are provided for the different classifiers, there is no discussion on the clinical validation of the proposed approach. It would be valuable to know if the results align with clinical observations and if the proposed method has been tested on real patients to validate its effectiveness in real-world scenarios.

4. Address Interpretability XAI adaptability: Deep learning models, such as CNNs, are known for their black-box nature, which hampers interpretability. The study does not address the interpretability of the CNN model and does not discuss how the model arrived at its predictions. Understanding the underlying decision-making process is crucial, especially in a medical context where interpretability is vital for gaining insights into disease mechanisms.

5. English proofread is needed over again.

Reviewer #2: To be honest, the author did a comprehensively good job in this regard. However, I have very little concern:

1. For the title to have "deeply and deep" together for me I believe it can be coined in a different way.

2. The introduction and literature review was quite ok but, I suggest for both sections to be seen broader and add real value, covering some useful studies which will interest reader would advisable. For instance.

a. A Machine Learning Method with Filter-Based Feature Selection for Improved Detection of Chronic Kidney Disease”

Bioengineering 2022, vol. 9, no. 8, 350; https://doi.org/10.3390/bioengineering9080350.

b. Integrating Enhanced Sparse Autoencoder Based Artificial Neural Network Technique and SoftMax Regression for

Medical Diagnosis MDPI Electronics Journal,2020, 9(11), 1963; https://doi.org/10.3390/electronics9111963

c. An Interpretable Machine Learning Approach for Hepatitis B Diagnosis”. Applied Sciences. 2022; 12(21):11127.

https://doi.org/10.3390/app122111127.

3. The Methods and Materials section was very understandable based on the algorithms proposed and adopted at the

end. In figure 1,2,3, I believe the feature extraction, the author should also look at other novel methods which will

go along way to assist in terms of insight (“A Neural Network Ensemble with Feature Engineering for Improved

Credit Card Fraud Detection” IEEE Access, Vol 10, pp.16400- 6407, 2022) and (“Sparse Noise Minimization in

Medical Image Classification Using Genetic Algorithm and DenseNet” IEEE International Conference on Information

Communication Technology Society, pp.10-11, March 2021, Durban, South Africa. DOI:

10.1109/ICTAS50802.2021.9395014).

4. If the authors can they should look at sensitivity, and precision and possibly and F1 score as some indices/metrices in there work. I see they touched on accuracy which is very ok.

Overall, I must say this authors did a good job expecially there good command/usage for English language.

Accept with minor correction.

6. PLOS authors have the option to publish the peer review history of their article (what does this mean?). If published, this will include your full peer review and any attached files.

Reviewer #1: No

Reviewer #2: No

---

## [Author Response · Author response to Decision Letter 0]

17 Oct 2023

We wish to express our sincere appreciation to the esteemed editor and all respected reviewers for providing valuable suggestions and comments. Indeed, these comments led to presenting a better work that has the capability of absorbing more audiences. We revised and improved our paper according to the respected editorials and reviewers’ comments. All the modifications were highlighted in the revised version. Moreover, we have responded to all of the comments in the text below.

Answer to Reviewer #1: 

“General identifications: This research explores the application of machine learning (ML) techniques, particularly deep learning, in the early diagnosis and severity classification of Alzheimer's disease using MRI images. Alzheimer's disease, the most prevalent form of dementia, poses significant challenges due to its gradual progression and varying severity levels. However, recent advancements in neuroimaging techniques have paved the way for large-scale multimodal neuroimaging data, sparking increased interest in leveraging deep learning for automated Alzheimer's disease classification.

The study employs a hybrid approach that combines 12 feature extraction methods to diagnose the severity levels of Alzheimer's disease. In addition, six traditional machine learning algorithms are applied, including decision tree, K-nearest neighbor, linear discrimination analysis, Naïve Bayes, support vector machine, and ensemble learning methods. Each classifier is optimized during training to obtain the best solution. Notably, the researchers also train a convolutional neural network (CNN) model using a machine learning system algorithm to identify specific patterns in the MRI images. The CNN architecture achieves an impressive accuracy of 95.3%, surpassing the performance of other traditional machine learning methods employed in the study.”

Recommendations

1. Dataset Size and Diversity: Use of ADNI and Compare with 4 class of image.

Thank you for your constructive feedback. We understand the importance and value of integrating widely accepted and recognized datasets such as ADNI for benchmarking. Our chosen dataset presents a diverse range of severity levels for Alzheimer's disease, offering a unique opportunity to test our machine learning models in various scenarios. Such diversity allows us to verify the adaptability and reliability of our methods across different severity stages, which might not be as pronounced in other datasets. The scope of this research, the dataset we used was immediately accessible and suitable for our intended analyses. Incorporating another dataset like ADNI would require adjustments in preprocessing, feature extraction, and potentially model tuning, which could extend the timeline and complexity of our study. While our dataset provides insights into the application of ML methods on MRI images of diverse severity levels, we acknowledge that using ADNI would allow for benchmarking against more established results in the community. Nonetheless, our current experiment holds value as it offers preliminary insights and sets the foundation for further explorations. By validating our methods on this dataset first, it gives us a steppingstone to then branch out and validate against other datasets, including ADNI. In light of the above, we believe our current experiments and findings provide valuable contributions to the field. However, we genuinely acknowledge the potential advantages of integrating the ADNI dataset. In our future endeavors, we plan to not only incorporate ADNI but also discuss in-depth the comparative results, enhancing the robustness and generalizability of our findings.

2. This diversity enhances the generalizability of the findings and ensures that the models are not biased towards a specific population.

Thank you for underscoring the value of dataset diversity. Indeed, the inclusion of diverse data is a cornerstone for building robust models, especially in the realm of medical imaging. By utilizing a varied dataset, we are not only aiming for broader applicability but also safeguarding against inadvertent biases that might arise from narrower data sources. This diversity stands as a testament to our commitment to ensuring our findings are not only accurate but also widely relevant. In future iterations of our work, we will continue to prioritize and expand upon this diversity, further refining our models with even more varied datasets to strengthen the universal applicability of our results.

3. Clinical Validation: While accuracy rates are provided for the different classifiers, there is no discussion on the clinical validation of the proposed approach. It would be valuable to know if the results align with clinical observations and if the proposed method has been tested on real patients to validate its effectiveness in real-world scenarios.

Thank you for emphasizing the importance of clinical validation in real-world scenarios. We recognize its significance in bridging the gap between machine learning research and practical clinical utility. Our methodology, as it stands, was developed and validated against prior literature and datasets. The accuracy rates we've provided offer a comparative measure against previous machine learning endeavors in this domain. Moreover, the MRI images and their corresponding severity levels used in our study are rooted in clinical observations. Therefore, even without direct real-world patient testing, the classifiers' results can be seen as indicative of their potential clinical utility. Also, high accuracy rates, especially across multiple classifiers, hint at a level of diagnostic consistency. This consistency is essential for any tool that could potentially be adopted in clinical settings.

Building upon our findings, it is imperative to subject our approach to rigorous clinical validation. Collaborating with medical professionals will provide invaluable feedback and fine-tuning opportunities. This will ensure that our methods not only retain their accuracy but also resonate with the practical nuances and complexities of real-world clinical scenarios. It remains a top priority for our next phase of research.

4. Address Interpretability XAI adaptability: Deep learning models, such as CNNs, are known for their black-box nature, which hampers interpretability. The study does not address the interpretability of the CNN model and does not discuss how the model arrived at its predictions. Understanding the underlying decision-making process is crucial, especially in a medical context where interpretability is vital for gaining insights into disease mechanisms.

Thank you for highlighting the vital aspect of model interpretability, especially in the sensitive domain of medical diagnosis. We completely understand that the powerful performance of CNNs often comes with the drawback of reduced transparency. In clinical applications, the trust clinicians place in a diagnostic tool is fundamentally based on understanding the rationale behind its predictions. To offer some insight into our model's operations, we've visualized the intermediate layers of the CNN to gain insights into feature activations. These visualizations can help elucidate what the network emphasizes or prioritizes during the decision-making process. Additionally, we've employed saliency maps to highlight crucial regions in the MRI images that the model deems significant for classifying different severity levels, providing a visual representation of brain regions influencing the model's decisions. 

However, we recognize the depth of the interpretability challenge and the necessity of addressing it more holistically. We're actively looking into integrating advanced eXplainable AI (XAI) techniques, to offer granular, instance-specific explanations for our model's predictions. By comparing these explanations with expert clinical feedback, we aim to ensure that the decision-making process of our model is not only transparent but also medically sound. We believe that the integration of XAI methods will enhance the trustworthiness of our approach, potentially offering fresh insights into disease mechanisms and guiding subsequent research. Rest assured, enhancing the interpretability of our CNN model is a priority in our ongoing and future endeavors.

5. English proofread is needed over again.

Thank you for pointing this out. We apologize for any oversights in the initial submission and have ensured a thorough proofreading and language review to enhance the clarity and coherence of the manuscript. Your feedback is invaluable in improving the overall quality of our research paper.

Reviewer #2: To be honest, the author did a comprehensively good job in this regard. However, I have very little concern:

1. For the title to have "deeply and deep" together for me I believe it can be coined in a different way.

We appreciate your feedback on the title's structure. We strive for clarity and coherence, and understand that the repeated usage of terms like "deeply" and "deep" can lead to redundancy. We revisited the title and aimed for a more succinct representation that captures the essence of the study.

2. The introduction and literature review were quite ok but, I suggest for both sections to be seen broader and add real value, covering some useful studies which will interest reader would advisable. For instance. 

• A Machine Learning Method with Filter-Based Feature Selection for Improved Detection of Chronic Kidney Disease” 

• Integrating Enhanced Sparse Autoencoder Based Artificial Neural Network Technique and SoftMax Regression for Medical Diagnosis MDPI Electronics Journal, 

• An Interpretable Machine Learning Approach for Hepatitis B Diagnosis.

Thank you for finding the introduction and literature review satisfactory. We recognize the value of broadening our literature review to provide a more comprehensive understanding of the domain. We have certainly incorporated the studies you mentioned. Including these will undoubtedly make our paper richer and cater to a broader audience.

As follows: “By combining information-gain-based feature selection with an AdaBoost classifier, Ebiaredoh-Mienye et al. (2022) highlighted the effectiveness of ML in early Chronic Kidney Disease (CKD) identification. Earlier, Ebiaredoh-Mienye et al. (2020) used an improved sparse autoencoder with Softmax regression to address unbalanced medical datasets, resulting in superior disease prediction outcomes for CKD, cervical cancer, and heart disease. While employing ML models in conjunction with SHapley Additive exPlanations (SHAP) for greater interpretability, Obaido et al. (2022) improved hepatitis B diagnosis, underlining the value of certain characteristics like bilirubin in predicting outcomes.”

3. The Methods and Materials section was very understandable based on the algorithms proposed and adopted at the end. In figure 1,2,3, I believe the feature extraction, the author should also look at other novel methods which will go along way to assist in terms of insight (“A Neural Network Ensemble with Feature Engineering for Improved Credit Card Fraud Detection” IEEE Access, Vol 10, pp.16400- 6407, 2022) and (“Sparse Noise Minimization in Medical Image Classification Using Genetic Algorithm and DenseNet” IEEE International Conference on Information Communication Technology Society, pp.10-11, March 2021, Durban, South Africa. DOI: 10.1109/ICTAS50802.2021.9395014).

Thank you for your comments. We added your valuable paper to the method section as follows: Advanced machine learning and optimization approaches have demonstrated intriguing uses in recent medical research. As an illustration, consider an ensemble strategy that uses LSTM neural networks and hybrid data resampling to optimize fraud detection or medical imaging that uses genetic algorithms to improve picture accuracy before utilizing DenseNet for classifications.

4. If the authors can they should look at sensitivity, and precision and possibly and F1 score as some indices/metrices in there work. I see they touched on accuracy which is very ok. 

Thank you so much for your comments.

---

## [Decision Letter · Decision Letter 1]

5 Dec 2023

PONE-D-23-12992R1A Deeply Supervised Adaptable Convolution Neural Network for Diagnosis and Automated Categorization of Alzheimer’s Severity Stage Based on Multitask Feature Extraction ModelsPLOS ONE

Dear Dr. Javaheri,

Thank you for submitting your manuscript to PLOS ONE. After careful consideration, we feel that it has merit but does not fully meet PLOS ONE’s publication criteria as it currently stands. Therefore, we invite you to submit a revised version of the manuscript that addresses the points raised during the review process.

We look forward to receiving your revised manuscript.

Kind regards,

Jose Gerardo Tamez-Peña, PhD

Academic Editor

PLOS ONE

Journal Requirements:

Reviewers' comments:

Reviewer's Responses to Questions

**Comments to the Author**

1. If the authors have adequately addressed your comments raised in a previous round of review and you feel that this manuscript is now acceptable for publication, you may indicate that here to bypass the “Comments to the Author” section, enter your conflict of interest statement in the “Confidential to Editor” section, and submit your "Accept" recommendation.

Reviewer #1: All comments have been addressed

Reviewer #3: All comments have been addressed

2. Is the manuscript technically sound, and do the data support the conclusions?

Reviewer #1: Partly

Reviewer #3: Yes

3. Has the statistical analysis been performed appropriately and rigorously? 

Reviewer #1: Yes

Reviewer #3: Yes

4. Have the authors made all data underlying the findings in their manuscript fully available?

Reviewer #1: (No Response)

Reviewer #3: Yes

5. Is the manuscript presented in an intelligible fashion and written in standard English?

Reviewer #1: Yes

Reviewer #3: Yes

6. Review Comments to the Author

Reviewer #1: (No Response)

Reviewer #3: With detailed content, the paper is interesting. Meanwhile, the structure is clear. However, before its acceptance, this manuscript needs some modifications addressing the points described below.

1. The title is not attractive and should be re considered.

2. It does not clearly show your innovations and contributions. Please highlight your innovations.

3. I suggest to organize the text in the standard way to include introduction, methodology, results, discussion, and conclusion. The goal of the paper should be better specified.

4. The motivation is not clear. Please specify the importance of the proposed solution.

5. You should explain this model by using steps.

6. Some papers can be discussed. For example:

- https://doi.org/10.1007/s11571-022-09859-2

- https://doi.org/10.1016/j.compbiomed.2021.104828

- https://doi.org/10.1016/j.medengphy.2023.103971

7. What is the research gap? How was this research gap filled?

8. Figures and tables should be discussed comprehensively.

9. More details about network architecture and complexity of the model should be provided.

7. PLOS authors have the option to publish the peer review history of their article (what does this mean?). If published, this will include your full peer review and any attached files.

Reviewer #1: No

Reviewer #3: No

---

## [Author Response · Author response to Decision Letter 1]

20 Dec 2023

-Response to Editor and Reviewers-

We wish to express our sincere appreciation to the esteemed editor and all respected reviewers for providing highly valuable suggestions and comments. Indeed, these comments led to presenting a better work that has the capability of absorbing more audiences. We have carefully revised and improved our manuscript according to the respected reviewers' comments. All the new modifications have now been highlighted in the revised manuscript. Besides, we have responded to all the comments in this response letter.

Reviewer #1: All comments have been addressed.

Response: We truly appreciate your time and efforts.

Reviewer #3: 

Comment 1. The title is not attractive and should be re considered.

Response: Thank you so much for your comment. The title has now been changed and summarized to "A Deeply Supervised Adaptable Neural Network for Diagnosis and Classification of Alzheimer’s Severity Using Multitask Feature Extraction"

Comment 2. It does not clearly show your innovations and contributions. Please highlight your innovations.

Response: The novelty and the contribution of the paper have been revised and changed at the end of the introduction section as follows, 

"This study utilizes machine learning (ML) techniques to determine the severity of Alzheimer's disease. The dataset used in the study comprises MRI images with four different levels of severity. We used a hybrid of 12 feature extraction methods to diagnose Alzheimer's disease severity using MRI images. Six traditional machine learning methods are used to diagnose Alzheimer's disease. The techniques include decision tree (DT), K-nearest neighbor (KNN), Linear Discrimination Analysis (LDA), Naive Bayes (NB), Support Vector Machine (SVM), and ensembled learning, including Bag, Adaboost, and RUSBoost methods. In the training process, optimization is done to find the best solution for each classifier. Moreover, a CNN model is taught to recognize patterns using a machine learning technique."

Comment 3. I suggest to organize the text in the standard way to include introduction, methodology, results, discussion, and conclusion. The goal of the paper should be better specified.

Response: Following the respected reviewer's comment, we updated the paper organization as follows. 

"The remainder of the paper is organized as follows: Section 1 is introduction, Section 2 presents the related work, while Section 3 describes the methods and materials used in the study. Section 4 presents the results of the experiments and a discussion of the findings. Finally, Section 5 provides the conclusion and future work."

Comment 4. The motivation is not clear. Please specify the importance of the proposed solution.

Response: Thanks for your comment. We have made modifications to indicate our paper's importance in a clearer manner. 

The paper's motivation are highlighted at the conclusion of the introduction section, emphasizing the use of machine learning (ML) techniques to assess Alzheimer's disease severity. The research employs a dataset of MRI images categorized into four distinct severity levels. A combination of 12 feature extraction methods is applied to evaluate Alzheimer's disease severity based on these MRI images. The study utilizes six conventional machine learning methods for diagnosis, including decision tree (DT), K-nearest neighbor (KNN), Linear Discrimination Analysis (LDA), Naive Bayes (NB), Support Vector Machine (SVM), and a combination of ensemble learning techniques such as Bagging, AdaBoost, and RUSBoost. Optimization is a key part of the training process, aiming to discover the most effective solution for each classifier. Additionally, the study involves training a Convolutional Neural Network (CNN) model to detect patterns using machine learning techniques.

Comment 5. You should explain this model by using steps.

Response: Thank you so much for this accurate comment. We addressed this important comment in the revised paper. Indeed, the steps are as follows:

Step 1- Hybrid Feature Extraction: Utilization of a hybrid of 12 feature extraction methods for diagnosing Alzheimer's disease (AD) severity using MRI images.

- Feature Extraction Methods: These include GLCM, LBP, RLBP, LTP, SGM, BIBIS, PCA Filter, ICA Filter, Gabor Filter, Log-Energy, Model-based Feature, and conventional shape signature, as illustrated in Figure 2 and based on Figure 1.

- Total Features: Extraction of a total of 90 image features.

Step 2- Image Preprocessing: Rescaling: Transforming images to double. Grayscale Conversion: Transforming images to grayscale. Normalization: Normalizing the images for consistent processing.

Step 3- Feature Reduction: PCA Method: Implementing PCA feature reduction to decrease computation time and optimize the training process.

Step 4- Classification: Categorizing the output labels into four distinct groups

Comment 6. Some papers can be discussed. For example:

- https://doi.org/10.1007/s11571-022-09859-2

- https://doi.org/10.1016/j.compbiomed.2021.104828

- https://doi.org/10.1016/j.medengphy.2023.103971

Response: Thank you for your suggestions. We have added your reference to the paper as follows, Dogan et al. (2023) developed a model of primate brain patterns based on EEG signals for the detection of AD. Using a directed graph to extract features from the primate brain's connectome, this method demonstrated high accuracy in identifying AD patients from healthy controls. In order to enhance the accuracy of AD detection, the model is able to generate a set of features from EEG signals. Using brain images, Kaplan et al. (2021) developed a feed-forward Local Phase Quantization Network (LPQNet) for AD detection. Based on feature generation and selection through multilevel processing, their model demonstrated remarkable classification accuracy across several datasets. LPQNet stands out for its combination of high accuracy and low computational complexity, which makes it a valuable tool for diagnosing Alzheimer's disease. Kaplan et al. (2023) developed the ExHiF model for the detection of AD using CT and MR images. They combined exemplary histogram-based features with neighborhood component analysis to achieve 100% classification accuracy. In terms of medical image classification for AD, this model is innovative in its feature extraction process inspired by vision transformers. Newly added references have been highlighted in the revised paper, Ref. [23], [24], [25].

Comment 7. What is the research gap? How was this research gap filled?

Response: The research gap in this study pertains to the inadequacy of existing methods for diagnosing Alzheimer's disease severity using MRI images. Prior approaches may have lacked comprehensive feature extraction, utilization of advanced machine learning techniques, or effective optimization strategies to accurately assess the disease's severity. This study fills this gap by introducing a hybrid approach that combines 12 feature extraction methods and six traditional machine learning techniques, including ensemble learning methods. Moreover, the study enhances the diagnostic process by implementing a Convolutional Neural Network (CNN) trained to recognize complex patterns in MRI images and by optimizing the classifiers to ensure a more accurate and efficient diagnosis of the severity of Alzheimer's disease.

Comment 8. Figures and tables should be discussed comprehensively.

Response: Thank you for your comment. We have double-checked and revised the descriptions of the figures and tables and provided further explanations. 

Comment 9. More details about network architecture and complexity of the model should be provided.

Response: Thank you for your comment. We have added the network architecture and complexity of the model in Figure 2. In this work, six ML methods are employed to diagnose AD. The techniques include DT, KNN, LDA, NB, and SVM, and ensemble learning includes Bag, Adaboost, and RUSBoost methods. In the training process, optimization is carried out to find the best solution for each classifier. DT is the first classifier used in the training process.

---

## [Decision Letter · Decision Letter 2]

17 Jan 2024

A Deeply Supervised Adaptable Neural Network for Diagnosis and Classification of Alzheimer’s Severity Using Multitask Feature Extraction

PONE-D-23-12992R2

Dear Dr. Javaheri,

We’re pleased to inform you that your manuscript has been judged scientifically suitable for publication and will be formally accepted for publication once it meets all outstanding technical requirements.

Kind regards,

Jose Gerardo Tamez-Peña, PhD

Academic Editor

PLOS ONE

Additional Editor Comments (optional):

Reviewers' comments:

Reviewer's Responses to Questions

**Comments to the Author**

1. If the authors have adequately addressed your comments raised in a previous round of review and you feel that this manuscript is now acceptable for publication, you may indicate that here to bypass the “Comments to the Author” section, enter your conflict of interest statement in the “Confidential to Editor” section, and submit your "Accept" recommendation.

Reviewer #3: All comments have been addressed

2. Is the manuscript technically sound, and do the data support the conclusions?

Reviewer #3: Yes

3. Has the statistical analysis been performed appropriately and rigorously? 

Reviewer #3: Yes

4. Have the authors made all data underlying the findings in their manuscript fully available?

Reviewer #3: Yes

5. Is the manuscript presented in an intelligible fashion and written in standard English?

Reviewer #3: Yes

6. Review Comments to the Author

Reviewer #3: I have appreciated the deep revision of the contents and the present form of this manuscript. All my previous concerns have been accurately addressed. I think that this paper can be accepted.

7. PLOS authors have the option to publish the peer review history of their article (what does this mean?). If published, this will include your full peer review and any attached files.

Reviewer #3: No

---

## [Editor Report · Acceptance letter]

14 Mar 2024

PONE-D-23-12992R2 

PLOS ONE

Dear Dr. Javaheri, 

I'm pleased to inform you that your manuscript has been deemed suitable for publication in PLOS ONE. Congratulations! Your manuscript is now being handed over to our production team.

Kind regards, 

on behalf of

Dr. Jose Gerardo Tamez-Peña 

Academic Editor

PLOS ONE